# Regulation of the Intestinal Stem Cell Pool and Proliferation in *Drosophila*

**DOI:** 10.3390/cells13221856

**Published:** 2024-11-08

**Authors:** Simona Trubin, Dhruv B. Patel, Aiguo Tian

**Affiliations:** 1Department of Biochemistry and Molecular Biology, Tulane University School of Medicine, Louisiana Cancer Research Center, New Orleans, LA 70112, USA; 2Tulane Aging Center, Tulane University School of Medicine, New Orleans, LA 70112, USA

**Keywords:** intestine, ISC pool, ISC proliferation

## Abstract

Understanding the regulation of somatic stem cells, both during homeostasis and in response to environmental challenges like injury, infection, chemical exposure, and nutritional changes, is critical because their dysregulation can result in tissue degeneration or tumorigenesis. The use of models such as the *Drosophila* and mammalian adult intestines offers valuable insights into tissue homeostasis and regeneration, advancing our knowledge of stem cell biology and cancer development. This review highlights significant findings from recent studies, unveiling the molecular mechanisms that govern self-renewal, proliferation, differentiation, and regeneration of intestinal stem cells (ISCs). These insights not only enhance our understanding of normal tissue maintenance but also provide critical perspectives on how ISC dysfunction can lead to pathological conditions such as colorectal cancer (CRC).

## 1. Background

Except for food digestion and nutrient absorption, the intestinal epithelium serves as a crucial barrier, protecting the internal gut milieu from external environmental threats. The *Drosophila* adult midgut, functionally equivalent to the mammalian small intestine, has become a valuable system for studying ISCs since their identification in 2006 [1,2,3,4,5,6].

During tissue homeostasis in the *Drosophila* intestine, the resident ISCs can self-renew and differentiate into committed cells (enteroblasts and pre-enteroendocrine cells) and then mature cells (enterocytes and enteroendocrine cells) to replace damaged cells (Figure 1A) [1,2,7]. In response to tissue damage, including injury and infection, ISCs, precursor cells, and mature cells can be damaged. ISCs then become transiently activated, increasing their proliferation and undergoing symmetrical divisions to produce more ISCs [1,2,7,8,9,10,11,12,13,14,15,16,17,18,19,20,21,22,23,24,25,26]. In addition, ISC progeny cells can also dedifferentiate to replenish the ISC pool [27,28,29].

Similarly, the mammalian intestinal epithelium is also a highly dynamic tissue, which is organized into crypts and villi. Crypt base columnar (CBC) cells, a subset of ISCs, reside between Paneth cells at the crypt base. These ISCs self-renew and differentiate into transit-amplifying (TA) cells, progenitor cells, and various mature cell types, such as goblet, tuft, and enteroendocrine cells, as well as enterocytes (Figure 1B) [30,31,32,33]. In response to dextran sodium sulfate (DSS) feeding, inflammation or ISC loss due to irradiation and ablation, secretory and enterocyte progenitors, Paneth cells, enteroendocrine cells, and tuft cells can undergo dedifferentiation to generate regenerative ISCs [34,35,36,37,38,39,40,41,42,43].

This review explores how ISC maintenance, division patterns, dedifferentiation, and proliferation are regulated during homeostasis and regeneration.

## 2. ISC Maintenance

Multiple signaling pathways have been found to regulate ISC maintenance (Figure 2 and Table 1). In *Drosophila*, Notch signaling plays a critical role in regulating ISC fate by inhibiting ISC self-renewal and promoting its differentiation [1,2,8]. Loss of Notch signaling leads to symmetric ISC divisions and the formation of intestinal tumors consisting of large clusters of ISCs and enteroendocrine-like (EE-like) cells. In contrast, activating Notch signaling induces premature differentiation of ISCs and precursor cells into mature enterocytes, resulting in the rapid loss of ISCs [1,2,8,44,45]. The Notch pathway activates the Enhancer of split complex (E(spl)-C), which inhibits the transcription factor Daughterless (Da) [44]. Additionally, Hairless regulates the ISC fate by repressing expression of E(spl)-C and reducing Notch activity [44]. But how asymmetric Notch signaling is established and maintained is not very clear. Recent studies found that Sara endosomes can determine the ISC fate through Notch signaling: during ISC mitosis, Sara endosomes are asymmetrically dispatched to the presumptive enteroblasts (EBs), which leads to Notch signaling bias by regulating Notch and Delta traffic [46]. Another study showed that the Par complex is asymmetrically segregated into apical daughter cells, promoting the EB fate by regulating Notch activity [24]. In addition, another study found that the basal daughter cell following the ISC division exhibits higher levels of BMP signaling than the apical cell, with BMP signaling promoting ISC fate by antagonizing Notch signaling [9]. In mammals, Notch signaling is required for the maintenance of the ISC pool, whereas Notch inhibition leads to reduced Lgr5^+^ cell numbers [47]. In addition, Notch signaling plays a crucial role in the differentiation of ISCs into absorptive enterocytes versus secretory lineages (goblet and enteroendocrine cells). Notch signaling favors the absorptive cell fate, while its inhibition promotes the secretory lineage [48,49,50].

The Wingless signaling pathway is one of key regulators of ISC maintenance. Loss of Wingless pathway components, like Frizzled or Armadillo, leads to a loss of ISCs and its overactivation produces extra ISCs [10]. Its ligand Wingless is expressed in the visceral muscle (Figure 3). Similarly, in mammals, Wingless signaling is essential for ISC maintenance [51]. Lgr5^+^ ISCs rely on Wnt signaling to maintain crypt integrity and prevent premature differentiation [30].

The Janus kinase-signal transducer and activator of transcription (JAK-STAT) pathway also plays a key role in ISC maintenance [52,53]. In *Drosophila*, the JAK-STAT pathway is activated by three ligands, Unpaired 1 (Upd1), Unpaired 2 (Upd2), and Unpaired 3 (Upd3), each with distinct functions. Upd1 is produced in precursor cells and is required throughout life to maintain basal turnover of the midgut epithelium by controlling ISC maintenance in an autocrine manner. Upd2 is produced in both precursor cells and mature enterocytes, playing a role in the regular maintenance of the gut. Upd3, on the other hand, is primarily upregulated in mature enterocytes during tissue damage, such as infection or inflammation. Its upregulation contributes to increased ISC proliferation in response to damage [53].

EGFR-Ras signaling plays a key role in maintaining ISCs (Figure 2). Its ligand Vein (Vn), specifically expressed in muscle cells, is important for ISC maintenance, while two additional ligands, Spitz and Keren (Krn), function redundantly to promote ISC maintenance [54] (Figure 3).

In terms of the interplay among Notch, Wingless, JAK-STAT, and EGFR in regulating ISC maintenance, previous studies indicate that Wingless, JAK-STAT, or EGFR signaling functions upstream of Notch [10,52,54]. Moreover, overactivated EGFR signaling could partially compensate the loss of Wingless or JAK/STAT signaling for maintaining ISCs [54].

In addition, the target of rapamycin complex (TORC), regulated by the tuberous sclerosis complex (TSC1/2), is essential for ISC maintenance [55]. Integrin serves as a regulator for the maintenance of ISCs [56]. The zinc-finger transcription factor *escargot* (*esg*) is required to prevent premature ISC differentiation [57]. Both GATAe and Forkhead (FoxA) act as transcription factors to regulate ISC homeostasis [58,59].

Together, these signaling pathways ensure the balance between ISC self-renewal and differentiation, maintaining intestinal tissue integrity and function during homeostasis and regeneration.

## 3. Division Patterns of Resident ISCs

During intestinal homeostasis, approximately 80% of ISCs undergo asymmetric cell division (ISC-EB) whereas only 20% undergo symmetric cell division (ISC-ISC, which generates two ISCs, or EB-EB, which leads to a loss of ISCs) [9,25,60,61] (Figure 4A). Thus, the balance of symmetric division and asymmetric division is crucial for maintaining the size of the ISC pool consistent.

The ISC division patterns are regulated by the Notch signaling. An excessive Notch signaling results in symmetric division for ISC differentiation, and less Notch signaling can induce symmetric division for ISC accumulation [60,61]. Other signaling pathways, including BMP, JNK, and insulin [9,25,62], also modulate ISC division patterns, with external factors such as bleomycin exposure and aging [15,62].

### 3.1. Asymmetric and Symmetric Division

In newly matured *Drosophila* adults, feeding induces an increase in ISC numbers and differentiated cells, mediated by insulin-like peptides (dILPs) from brain neurons. Feeding and fasting/refeeding trigger dILP3 expression (Figure 3), which activates insulin signaling in ISCs, promoting symmetric division (ISC-ISC) and increasing the ISC pool [25] (Figure 4A). However, the exact mechanism by which insulin signaling regulates ISC division modes remains unclear.

The Bone Morphogenetic Protein (BMP) pathway is another regulator for the ISC division pattern. BMP signaling is asymmetrically distributed between ISCs and EBs, with higher activity in ISCs [9]. Loss of BMP signaling leads to a loss of ISCs due to increased symmetric division into EBs. Conversely, overexpression of BMP ligands Dpp and Gbb or a constitutively active form of type I receptor *Tkv* promotes symmetric division towards ISCs, expanding the ISC pool [9] (Figure 4A). In this process, BMP signaling acts upstream of Notch signaling, but the precise relationship between the two pathways remains to be elucidated.

Additionally, the Par complex also regulates the ISC division pattern [24] (Figure 4A). Loss of the Par complex leads to an increase in ISC numbers, as it promotes symmetric ISC-ISC divisions. Conversely, the gain of function of the Par complex drives EB-EB divisions through Notch pathway activation.

In response to tissue damage, such as bleomycin feeding, ISC numbers increase through symmetric division, driven by upregulated BMP signaling in ISCs through upregulation of Dpp and Gbb in ECs (Figure 3) [15]. The expansion of the ISC pool is tightly regulated; elevated BMP signaling in enterocytes (ECs) suppresses Dpp expression and reduces BMP signaling in ISCs, thereby limiting ISC pool expansion [15].

**Figure 4 cells-13-01856-f004:**
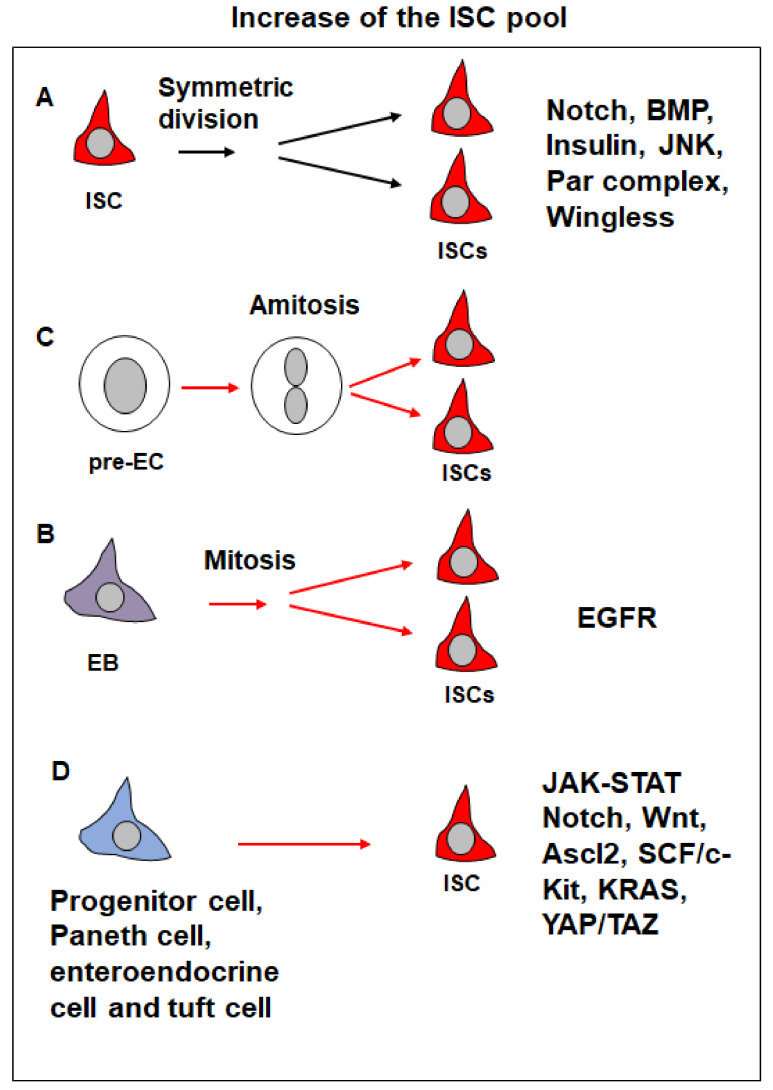
Increase of the ISC pool with the symmetric division of ISCs, mitosis of EBs, amitosis of pre-ECs, and dedifferentiation of committed cells. (**A**) The symmetric division of an ISC generates two ISCs. The regulatory pathways include Notch, BMP, insulin, and JNK. (**B**) The EB re-enters mitosis to generate two ISCs, which is regulated by EGFR signaling. (**C**) The pre-EC generates two ISCs through amitosis. (**D**) Dedifferentiation of committed cells such as progenitor cells, Paneth cells, enteroendocrine cells, and tuft cells into ISCs. The signaling involved includes JAK-STAT, Notch, Wnt, Ascl2, SCF/c-Kit, KRAS and YAP/TAZ.

In the developing mouse intestine, Lgr5^+^ ISCs initially undergo symmetric divisions to expand the ISC pool. Once the pool reaches sufficient size, divisions shift toward asymmetric division, generating both ISCs and transient amplifying (TA) progenitors [63,64]. However, in the adult mouse small intestine, most Lgr5^+^ ISC divisions remain symmetric, giving rise to either two ISCs or two TA cells [65,66]. More recent studies suggest that under homeostatic conditions, asymmetric division becomes predominant, with approximately 60% of ISC divisions being asymmetric [67].

### 3.2. Mitotic Spindle Orientation

The orientation of the mitotic spindle plays a critical role in determining whether stem cell divisions are symmetric or asymmetric. In the *Drosophila* intestine, the activity of JNK and Kif1a levels directly influence the plane of the mitotic spindle, dictating whether daughter cells will be symmetric or asymmetric. Overactivation of JNK signaling results in planar spindle orientation and increased symmetric division, while elevated Kif1a expression promotes oblique spindles and asymmetric division [62].

In the mouse intestine, ISC spindle orientation also contributes to asymmetric/symmetric division [68]. Cells located in the 1–7 cell region of the crypt orient their spindles perpendicular to the apical lumen, promoting asymmetric division, while cells in the >7 region align their spindles parallel to the lumen, resulting in more symmetric divisions [69]. Disruption of this orientation, such as in *Adenomatous polyposis* coli (*Apc*) mutants, is associated with CRC initiation, highlighting the importance of spindle orientation in maintaining tissue integrity [70].

## 4. Regeneration of ISCs

During homeostasis, resident ISCs self-renew to maintain tissue balance. However, in response to the loss of resident ISCs, such as in the mammalian intestine, ISC progeny cells can dedifferentiate to produce regenerative ISCs. In contrast, earlier studies suggested that in the *Drosophila* intestine, progenitor cells and mature cells cannot dedifferentiate into ISCs once precursor cells are lost [71]. Recent findings, however, challenge this notion, revealing distinct regenerative processes occurring in the *Drosophila* intestine.

### 4.1. Regeneration of ISCs Through Amitosis During the Starvation-Refeeding Process

Under normal conditions, ISCs continuously self-renew to maintain intestinal homeostasis. However, stress conditions such as starvation can lead to ISC loss, posing a challenge to intestinal integrity. During refeeding, the ISC pool is replenished through a regenerative process driven by amitosis—a form of cell division distinct from conventional mitosis [27] (Figure 4B).

In this process, tetraploid pre-enterocytes divide without forming a mitotic spindle, resulting in binuclear cells. These cells can subsequently divide to form two daughter cells, which acquire ISC identity, restoring the ISC population. Amitosis serves as a rapid response to stress-induced ISC loss, but it also carries risks, such as random chromosome segregation, potentially leading to genetic instability.

The discovery of amitosis in ISC regeneration highlights both the resilience and potential vulnerability of the intestinal epithelium during physiological stress. Understanding this balance between rapid regeneration and genetic risks could shed light on intestinal disorders and cancers where aberrant ISC behavior is implicated.

### 4.2. Regeneration of ISCs Through Mitosis upon Bacterial Infection

The *Drosophila* midgut, a well-characterized ISC model, reveals a fascinating mechanism of ISC regeneration in response to bacterial infection. Normally, ISCs produce EBs, which differentiate into ECs after losing their proliferative capacity. However, bacterial infections or the activation of EGFR-Ras signaling can induce EBs to revert to a proliferative state, allowing them to re-enter the mitotic cycle [28,72] (Figure 4C and Table 1).

Upon infection, EGFR-Ras signaling is activated in EBs through ligands such as Vein (Vn) and Keren (Krn). Although these proliferating EBs initially lack ISC markers, they regain multipotency after mitosis, functioning as ISCs and expanding the ISC pool [28]. This process ensures robust regenerative responses to epithelial damage caused by bacterial infection, underscoring the plasticity of the intestinal epithelium.

This finding reveals an alternative regeneration mechanism beyond direct reprogramming. The ability of EBs to regenerate ISCs via mitosis illustrates the adaptability of ISCs in response to environmental challenges and offers potential insights into tissue regeneration therapies.

### 4.3. Nutrient-Driven Dedifferentiation of EE Cells into ISCs

Nutrient availability plays a critical role in the adaptability of the adult intestine, influencing ISC pool size through processes like dedifferentiation. Recent studies have shown that during nutrient abundance, EE cells in the *Drosophila* intestine can dedifferentiate into ISCs, particularly following refeeding after starvation through the JAK-STAT signaling pathway [29]. Specifically, upregulation of *upd2* and *upd3* by nutrition fluctuation can activate JAK-STAT signaling in EE cells to induce EE cells to dedifferentiate into ISCs.

EE cell dedifferentiation is nutrient dependent. In flies deprived of sucrose or amino acids, dedifferentiation is significantly reduced, indicating that these nutrients are essential for the process. Although dedifferentiation occurs at lower frequencies when only sucrose and amino acids are present, a nutrient-complete diet enhances this mechanism.

This nutrient-driven dedifferentiation underscores the plasticity of the intestinal epithelium and the critical role of diet in regulating stem cell dynamics. Understanding these nutrient-dependent mechanisms may provide insights into improving tissue regeneration and repair strategies.

### 4.4. Regeneration of ISCs in Mammalian Intestines

In mammals, ISCs can be regenerated from various ISC progeny cell types after damage induced by DSS feeding [39,73] or resident ISC ablation [35,36,37,38,74,75,76,77] (Figure 4D). Dll1^+^ secretory progenitor cells can form organoids that contain Lgr5^+^ cells when exposed to excess WNT3A in vitro, demonstrating their regenerative potential [36]. In vivo, these Dll1^+^ cells can regenerate fully functional Lgr5+ stem cells after ISC ablation. Similarly, secretory progenitors marked by Atoh1 expression can contribute to ISC regeneration. Lineage tracing of Atoh1^+^ cells confirm their ability to repopulate crypts following damage from irradiation, ISC ablation, or DSS treatment [40,41]. Paneth cells and their precursors also exhibit dedifferentiation potential after injury, as demonstrated by lineage tracing experiments following irradiation [38], DSS treatment [39], or doxorubicin-induced damage [37]. Alkaline phosphatase (ALPI)-positive enterocyte progenitors can similarly dedifferentiate into Lgr5^+^ ISCs after ablation of ISCs [35]. Additionally, BMI1^+^ Prox1+ enteroendocrine cells can acquire stemness during injury-induced regeneration [77]. Finally, tuft cells can acquire ISC properties under homeostatic conditions and played a critical role in regeneration [78].

Several signaling pathways, including Notch [37,38], SCF/c-Kit [39,79], Wnt [42], *Ascl2* [34], *KRAS* [42], and YAP/TAZ [80], play a critical role in this process (Figure 4D and Table 1). For instance, the transcription factor *Ascl2* plays a critical role in maintaining Lgr5^+^ ISCs, with its expression being restricted to these stem cells [34]. Although *Ascl2* is not essential for normal intestinal homeostasis, its absence significantly impairs the ability to restore the Lgr5^+^ ISC pool after injury, such as irradiation or ablation. Following ISC loss, *Ascl2* expression is activated in cells from the middle crypt region, driving them to dedifferentiate and repopulate the ISC pool at the base of the crypt [34].

The generation of ISCs from their progeny cells through dedifferentiation provides promising insights for tissue regeneration therapies. Multipotent mesenchymal stem cells (MSCs) have long been studied for their therapeutic potential due to their ability to promote tissue repair, reduce inflammation, and improve outcomes across diverse animal disease models [81]. This concept has shown practical success in Crohn’s disease (CD), where stem cell therapy has proven highly effective for severe CD phenotypes with minimal adverse effects [82]. Similar to MSCs, ISCs are multipotent and promote tissue healing, suggesting potential therapeutic applications for intestinal damage caused by disease or other stressors. Studies in mammals also reveal that ISC progeny cells can dedifferentiate and initiate tumor formation [42,78], underscoring both the therapeutic promise and the need for careful exploration of ISC-based therapies.

## 5. ISC Proliferation

The ISC proliferation is regulated by multiple signaling pathways in *Drosophila* and mammalian intestines (Figure 2 and Figure 3 and Table 1).

### 5.1. EGFR and JAK-STAT Signaling

In the adult *Drosophila* midgut, the EGFR and JAK-STAT signaling pathways are the primary regulators of ISC proliferation, playing essential roles during both homeostasis and regeneration (Figure 2) [11,21,22,52,54,83,84,85,86,87,88,89,90,91,92,93]. In detail, ectopic activation of either pathway induces dramatic ISC proliferation and even hyperplasia [11,22].

Both pathways are activated in the precursor cells by several functionally redundant ligands or cytokines. The EGFR ligands include Vn, Krn, and Spitz, which are expressed in the visceral muscle cells (VMs), enterocytes, and precursor cells, respectively [11,54] (Figure 3). Expression of the EGFR ligands is induced by damage or stress [11,21,84]. The *Drosophila* cytokines for the JAK-STAT pathway include Upd1, Upd2, and Upd3 [22,53]. The expression of these *upds* is regulated by the EGFR pathways [11]. For example: bacterial infection can activate EGFR-Ras signaling in precursor cells, inducing *upd* expression, while activation of EGFR-Ras signaling in ECs upregulates *upd3* expression [11] (Figure 3). Additionally, JAK-STAT signaling activation in muscle cells can promote expression of *Vn* [21] (Figure 3). The *upd* cytokines are also regulated by multiple signaling pathways, including Hippo [18,19,20], Misspahen-Warts-Yki/Sd [94], Hedgehog [16], JNK [11,22], Bursicon-DLGR1 [95], BMP [93,96], Notch [85], the AP-1 complex (D-Jun and D-Fos), and the Src-MAPK [11,93]. Therefore, these cytokines are upregulated to activate JAK-STAT signaling in ISCs to promote ISC proliferation.

In mammals, EGFR signaling is crucial for promoting epithelial cell survival, migration, and proliferation during tissue repair. Activation of EGFR stimulates downstream pathways such as RAS-RAF-MEK-ERK and PI3K-AKT, which are essential for cell growth and preventing apoptosis. Studies show that loss of EGFR activity induces quiescence of ISCs [97], while hyperactivation can lead to hyperplasia and intestinal polyps, which are precursors to CRC [98,99].

The JAK-STAT pathway is integral for regulating the immune response within the intestine. Cytokines like IL-6, IL-22, and IL-10 activate JAKs, which then phosphorylate Signal Transducer and Activator of Transcription (STAT) proteins. In the context of intestinal injury, STAT3 activation is particularly important for promoting intestinal regeneration and barrier restoration. However, persistent activation of JAK-STAT signaling, particularly STAT3, has been linked to chronic inflammation and the progression of CRC, as it drives uncontrolled proliferation and tumor formation [100,101]. In addition, JAK/STAT-1 signaling regulates activation of reserve ISCs in response to acute intestinal inflammation [102].

The cross-talk between these pathways is critical in ensuring proper control of cell proliferation, and this cross-regulation amplifies the cellular response to stress and injury, but when dysregulated, it contributes to tumorigenesis and cancer metastasis.

### 5.2. The Hippo Pathway

Recent studies have found that the Hippo pathway acts as a negative regulator of ISC proliferation in *Drosophila* [17,18,19,20,94,103]. Specifically, the loss of Hippo signaling components, such as Hippo and Warts (Wts), coupled with the activation of the transcriptional co-activator Yorkie (Yki), has been shown to stimulate ISC proliferation through the JAK-STAT signaling pathway in precursor cells or enterocytes (Figure 2 and Table 1) [18,19,20]. The activation of Yki or knock-down of Hippo in precursor cells can promote ISC proliferation by upregulating upd expression ISC [20], additionally, the activation of Yki or knock-down of Hippo in ECs can promote ISC proliferation by inducing *upd3* [18,19] or *upd* [52] expression. Moreover, Misshapen—a member of the germinal center protein kinase family—can interact with Wts to suppress Yki expression in EBs, which regulate *upd3* expression [94].

This pathway is highly conserved in mammals, where its homolog, YAP, has been implicated in promoting organ growth and tissue regeneration [104]. Dysregulation of Hippo signaling in mammals can lead to excessive ISC proliferation [105,106].

### 5.3. JNK Signaling

The JNK pathway plays a vital role in regulating ISC proliferation in response to various stressors, such as bacterial infections, DSS-induced damage, and aging (Figure 2 and Table 1). In *Drosophila*, JNK signaling is rapidly activated when the intestine encounters damage or stress. Once activated, JNK induces the expression of key signaling ligands (cytokines) such as *upd3*, *vn*, *spi*, and *krn* for activating the JAK-STAT and EGFR pathways, which are known to drive ISC proliferation and tissue repair [11,22,23,93].

Upon intestinal injury or infection, JNK signaling serves as an immediate defense mechanism by initiating the regenerative response. For example, DSS-induced damage to the intestinal epithelium leads to JNK activation in EBs, resulting in the activation of Hedgehog (Hh) signaling. Activated Hh signaling can upregulate *upd2* expression in EBs, which stimulates the JAK-STAT pathway in ISCs [15] (Figure 2 and Figure 3). In ECs, activated JNK can upregulate *upd* expression to activate JAK-STAT signaling in ISCs [22] (Figure 3). Additionally, aging can also activate JNK signaling, leading to increased ISC proliferation but a decline in differentiation [23].

In mammals, JNK signaling also plays an important role in the regulation of intestinal homeostasis and response to injury. Its activation is sufficient to induce cell proliferation in the intestinal crypt and increases tumor incidence and tumor growth in an inflammation-induced CRC model [107].

### 5.4. Insulin Signaling

The insulin signaling pathway is a conserved nutrient-sensing regulator that plays a critical role in regulating both ISC proliferation and symmetric division. In *Drosophila*, studies have shown that activation of insulin signaling within precursor cells promotes ISC proliferation, while the loss of insulin signaling inhibits ISC proliferation in response to tissue damage, such as that caused by DSS or bleomycin [108]. This suggests that insulin signaling is essential for ISC regeneration following injury (Figure 2). Notably, insulin signaling influences ISC proliferation both autonomously, within the ISCs themselves, and non-autonomously, by modulating the surrounding signaling [109].

In mammals, the relationship between insulin signaling and ISC proliferation is more pronounced. For example, long-term calorie restriction in mice has been shown to shorten intestinal villi and decrease the number of differentiated epithelial cells, while increasing the ISC population through non-autonomous mechanisms, specifically by inhibiting the mammalian target of rapamycin complex 1 (mTORC1) in Paneth cells [110,111]. Interestingly, while calorie restriction expands the ISC population in mice, it contrasts with *Drosophila*, where long-term nutrient deprivation reduces ISC divisions [6].

### 5.5. Hh Signaling

The Hh signaling pathway is a crucial regulator of tissue homeostasis and regeneration, functioning in various developmental processes [112]. In *Drosophila* midguts, recent studies have revealed that Hh signaling plays a role in regulating ISC proliferation [16,113]. While the loss of Hh signaling in ISCs and their lineages does not disrupt homeostasis, activation of this pathway, either through overexpression of the active form of Ci or the inactivation of Hh signaling inhibitors such as *patched* (*ptc*) or Debra, induces ISC proliferation. This indicates that Hh signaling is not necessary for ISC proliferation under normal conditions but becomes critical in response to tissue damage, such as that caused by DSS [16]. Upon such damage, the Hh ligand is upregulated in precursor cells via the JNK pathway, which in turn activates Hh signaling in both ISCs and EBs. Notably, Hh signaling in EBs is essential for DSS-induced ISC proliferation. Activation of Hh signaling in EBs leads to the upregulation of ligands such as *upd2*, and blocking this upregulation can reduce ISC proliferation triggered by DSS or Hh signaling activation (Figure 2 and Figure 3). Additionally, Hh signaling becomes activated in the aged intestine, and inhibiting this pathway can reduce aging-induced phenotypes [113].

In mammals, Hh signaling inhibits proliferation while promoting the differentiation of the gastric epithelium [114]. Conditional removal of Ptch1, a key inhibitor of the Hh pathway, results in the depletion of the epithelial precursor cell pool due to premature differentiation, further emphasizing the pathway’s role in maintaining the balance between ISC proliferation and ISC differentiation in tissue regeneration [115,116].

### 5.6. BMP Signaling

BMP signaling is a regulator of ISC proliferation and differentiation [96,117] (Figure 2). Studies have shown that the loss of BMP signaling in ECs leads to widespread EC death, triggering the secretion of JAK-STAT cytokines (*upd*, *upd2* and *upd3*) and EGFR signaling ligands from the dying cells, which in turn causes a non-autonomous increase in ISC proliferation (*vn*, *spi* and *krn*) [96]. In contrast, Guo et al. [117] reported that injury-induced BMP signaling acts autonomously in ISCs to limit their proliferation (Figure 2). This regulation is complex, with EGFR signaling required for the ISC proliferation that results from BMP signaling loss, while JAK-STAT signaling operates upstream to modulate BMP signaling.

However, the role of BMP signaling in ISC regulation is not entirely clear cut. Clonal analyses of mutations in BMP pathway components have yielded varying results. Some studies report increased clone size, others show no significant change, and some observe reduced clone size coupled with ISC loss [9,117,118]. These discrepancies may be explained by the extent of BMP pathway inactivation, where a partial loss of BMP signaling leads to ISC over-proliferation, while a more complete loss results in ISC depletion.

In mammals, BMP signaling counteracts the proliferative signals of the ISC niche, functioning to restrict proliferation. Inhibition of BMP signaling, such as by expressing Noggin (a BMP antagonist) or Bmpr1a deletion, results in crypt hyperplasia, development of ectopic crypts, and disrupted villus morphogenesis [119,120].

### 5.7. Other Signaling Pathways and Transcription Factors

Several additional signaling pathways play important roles in regulating ISC proliferation (Figure 2). The PDGF/VEGF signaling pathway is one such regulator. In *Drosophila*, PVR, a receptor homologous to mammalian PDGF and VEGF receptors, is activated in ISCs and is required for maintaining ISC proliferation during homeostasis [121] (Figure 2A). Another significant pathway is the Wingless pathway. Activation of this pathway in progenitor cells drives ISC over-proliferation [122,123], while activation in ECs inhibits the proliferation of neighboring ISCs, suggesting a context-dependent effect [124].

Mammalian leucine-rich repeat-containing G protein-coupled receptors (LGRs) have emerged as key regulators in stem cells, with LGR5, for instance, serving as a stem cell marker in the intestine [125]. In *Drosophila*, DLGR2, the ortholog of mammalian leucine-rich repeat-containing G protein-coupled receptors (LGR4-6), along with its ligand Bursicon, can negatively regulate ISC proliferation in muscle cells through the regulation of *vn* [95].

Regarding the upregulation of *upd3*, the study showed that the AP-1 complex (D-Jun and D-Fos) can regulate *upd3* expression and the upstream signaling of the AP-1 complex includes JNK signaling, P38 kinases, and the MAPK/D-ERK pathway [93]. In addition, the Src/Raf/Dsor1/MAPK pathway is involved in regulating *upd3* expression [93], while Notch appears to suppress *upd* expression [85].

The Sox family of transcription factors also has important roles in stem cell regulation (Figure 2A) [126,127]. In *Drosophila*, Sox21a, a member of the Sox B gene family, is expressed in both ISCs and EBs. In ISCs, it is critical for promoting proliferation under both normal and stress conditions [128]. Sox21a is regulated through the Ras/Erk and JNK pathways via Fos, and its activation is essential for ISC proliferation in response to these signals [128]. In EBs, Sox21a contributes to EC differentiation, and its loss leads to increased expression of mitogens such as *spi*, which promotes ISC proliferation through paracrine signaling [129,130].

Additionally, Sox100B, the *Drosophila* ortholog of mammalian Sox9, integrates signals from the JAK/STAT and EGFR pathways to regulate ISC proliferation and differentiation, underscoring its role in maintaining intestinal homeostasis [131].

In mammals, the role of Sox genes in ISC regulation is similarly important. Sox9, in particular, is critical for ISC proliferation and differentiation in the intestinal crypts [132,133]. However, the exact role of Sox9 as either an oncogene or tumor suppressor in intestinal cancers remains debated [134]. Other Sox genes, such as Sox2, Sox4, and Sox17, also contribute to ISC proliferation and differentiation in the mammalian intestine [135,136,137].

While these discoveries have expanded our understanding of ISC regulation, many questions remain. For instance, how these pathways are differentially activated under conditions like aging or bacterial infection is still unclear. Both aging and bacterial infections are known to activate EGFR and JAK-STAT signaling, but the molecular mechanisms underlying these responses are poorly understood. Future research is necessary to uncover how these pathways influence ISC behavior and tissue homeostasis under varying physiological stresses.

## 6. Summary and Future Perspectives

This review discussed and summarized the various mechanisms by which the ISC pool and proliferation are regulated. During homeostasis, resident ISCs self-renew and differentiate to replace damaged cells. ISC maintenance under normal conditions is primarily regulated by pathways including Notch, Wingless, JAK-STAT, EGFR, Par complex, and BMP. Notch signaling promotes ISC differentiation into absorptive enterocytes, while its inhibition directs differentiation into secretory cells. Studies indicate that Wingless, JAK-STAT, EGFR, and the Par complex function upstream of Notch, whereas BMP acts downstream. Yet, the precise molecular mechanisms within these regulations remain to be clarified.

Regarding the regulation of the ISC pool size, symmetric division of ISCs can either expand or reduce the pool size depending on whether ISCs or EBs are produced. This balance is influenced by insulin, BMP, Par complex, and JNK signaling. The anti-cancer drug bleomycin has also been shown to expand the ISC pool via BMP signaling by upregulating Dpp and Gbb. Additionally, ISC regeneration expands the ISC pool through mechanisms such as EC amitosis, EB mitosis, and direct reprogramming of ISC progeny cells. Production of regenerative ISC is mediated by factors including DSS administration [39,73], ISC ablation [35,36,37,38,74,75,76,77], nutrient fluctuations [27,29], and pathogenic bacterial infection [28]. Although amitosis replenishes the ISC pool, random chromosome segregation may result in genetic instability and cancer, limiting its therapeutic application. Most studies have so far focused on dedifferentiation following ISC ablation; future studies should explore dedifferentiation in response to physiological damage and investigate the mechanisms by which ISC progeny acquire ISC fate.

In terms of ISC proliferation, conserved signaling pathways interact together in response to tissue damage. For example, bacterial infection can activate EGFR, Hippo, and JNK pathways to upregulate expression of *upds* and activate JAK-STAT signaling in ISCs, promoting ISCs proliferation. Specific pathways respond differently to various forms of tissue damage; for example, Hedgehog signaling is activated by DSS but not by bleomycin [16], whereas BMP signaling is activated by bleomycin but not DSS [15]. However, the molecular mechanisms remain to be explored.

Collectively, these pathways maintain intestinal tissue integrity and function during homeostasis and regeneration. Disruption of specific pathways can lead to tumor growth—for example, Notch pathway loss can induce ISC tumors [45]. Insights into these mechanisms may aid in understanding gastrointestinal diseases and CRC pathogenesis [138]. Advances in intestinal regeneration suggest the importance of cellular plasticity and dedifferentiation in injury-induced regeneration, often involving upregulated stem cell niche signaling. Understanding this regenerative hierarchy may lead to improved treatments for chronic intestinal injuries like inflammatory bowel disease and novel cancer therapies targeting stem cells and the tumor microenvironment. In mammalian studies, CRC development has been linked to activation of the Wingless pathway, KRAS signaling, and other oncogenic changes. Based on these knowledges, the monoclonal antibody targeting EGFR has become essential components in the treatment of patients with metastatic CRC [139].

Much of this research has relied on *Drosophila* and mouse models, which offer insights but have limited direct applicability to human intestinal disease. The development of human organoids from normal and diseased epithelia [140] presents a promising platform to study cell fate mechanisms and lineage specification. Organoids derived from human tissues enable investigation of differentiation potential across tumorigenic stages and within genetically normal cells. Future research will likely focus on the factors promoting stem cell activity in regenerative intestinal cells, bringing the field closer to clinical applications that may offer new therapeutic avenues for gastrointestinal diseases and CRC.

## Figures and Tables

**Figure 1 cells-13-01856-f001:**
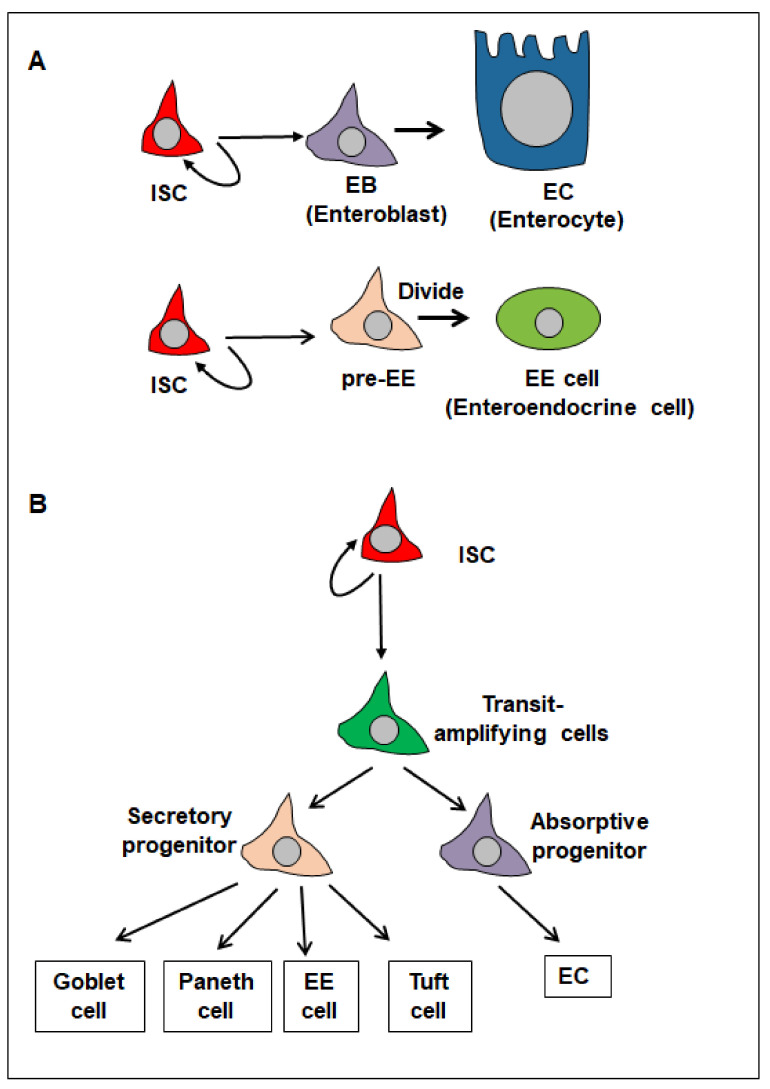
ISC lineages in *Drosophila* and mammalian adult intestines. (**A**) *Drosophila* ISC lineages include EBs (enteroblasts), pre-EE cells (pre-enteroendocrine cells), ECs (enterocytes), and EE cells (enteroendocrine cells). (**B**) Mammalian ISC lineages include Transit-amplifying cells, Secretory progenitors, Absorptive progenitors, ECs, EE cells, Paneth cells, goblet cells, and tuft cells.

**Figure 2 cells-13-01856-f002:**
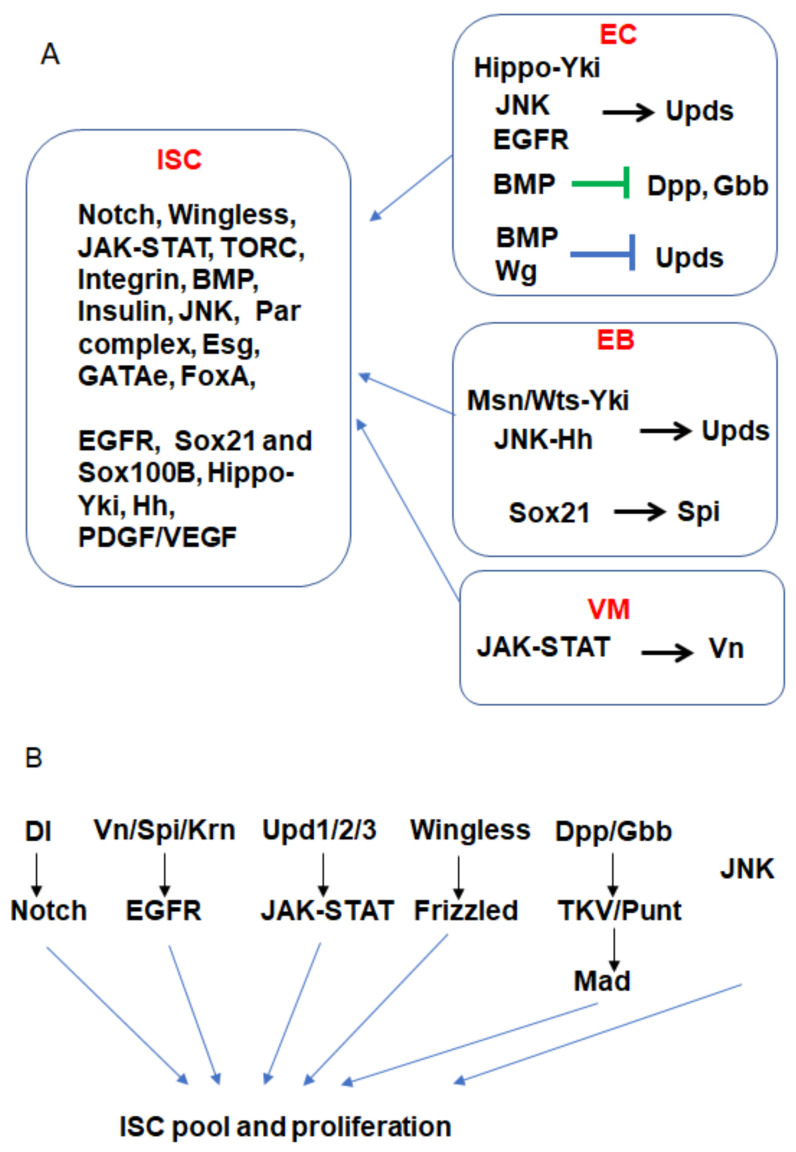
Signaling networks in regulation of ISCs. (**A**) The signaling pathways and transcription factors from ISCs, EBs, ECs and VM regulate ISC maintenance, fate determination, and proliferation. (**B**) The signaling pathways regulate ISC pools and proliferation in *Drosophila*.

**Figure 3 cells-13-01856-f003:**
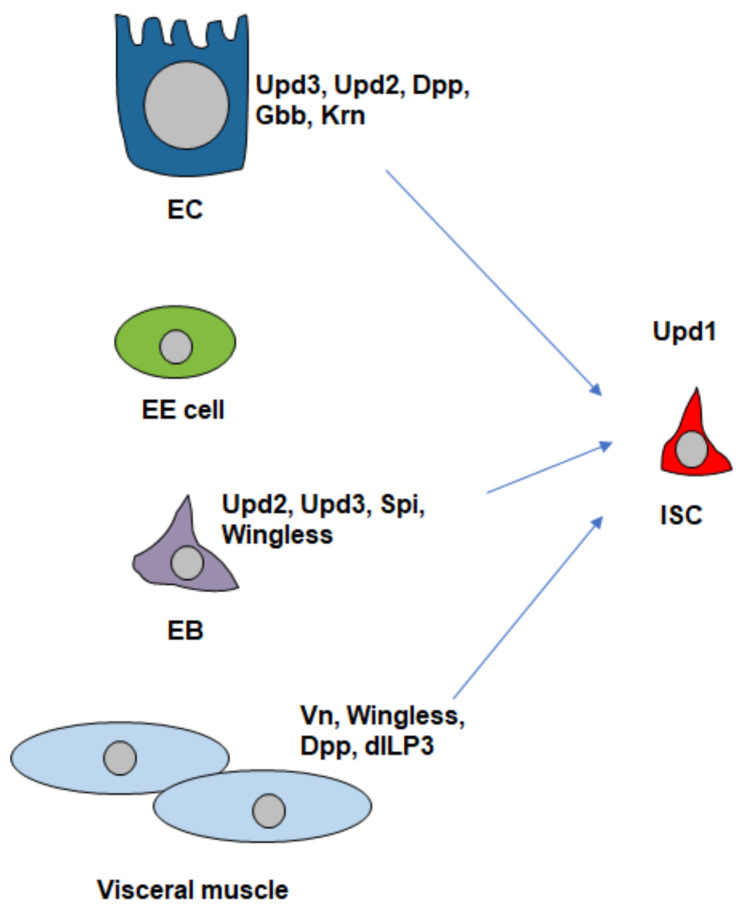
Ligands and cytokines produced in ISCs, EBs, ECs, and visceral muscle activate JAK-STAT, EGFR, Wingless, BMP, and insulin signaling pathways to regulate ISC maintenance and proliferation.

**Table 1 cells-13-01856-t001:** Comparison of signaling network that regulates ISC pools, ISC proliferation, and ISC regeneration in the *Drosophila* and mammalian intestines.

	*Drosophila*	Mammals
ISC pool	Notch, Wingless, JAK-STAT, TORC, Integrin, BMP, Insulin, JNK, Par complex, Esg, GATAe, FoxA,	Notch, Wingless, BMP
ISC proliferation	EGFR, JAK-STAT, JNK, Notch, Hh, Wingless, BMP, Insulin, Hippo, Sox21 and Sox100B.	EGFR, JAK-STAT, JNK, Hh, BMP, Insulin, Hippo, Wingless.
ISC regeneration	EGFR-Ras, JAK-STAT	Notch, SCF/c-Kit, Wingless, Ascl2, KRAS, YAP/TAZ

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
