# Peer review of "Regulation of the Intestinal Stem Cell Pool and Proliferation in Drosophila"

_cells, 2024, doi:10.3390/cells13221856_

Round 1
Reviewer 1 Report
Comments and Suggestions for Authors
Comments are provided in the attachment.

Author Response
Comments/Suggestions
- Line 43: The review explores multiple signaling pathways, however, the cross-talk and hierarchical relationships between key regulators, such as BMP and Notch, remain unclear. More detailed explanations of how these pathways interact under different stress conditions would improve the understanding of ISC regulation.
We agree with the reviewer’s suggestion and have added further details on the cross-talk between Notch and the Par complex (or BMP, EGFR, Wingless, and JAK-STAT) in P3, between EGFR and JAK-STAT in P11, and between Hh and JAK-STAT in P12.
- Line 133: In line text reference format.
We thank the reviewer for pointing that out and revised it.
- Though various regeneration mechanisms were discussed, some processes, like the dedifferentiation of certain cells into ISCs, are not fully explained. Further mechanistic details are needed to clarify how these processes contribute to tissue homeostasis and cancer prevention.
Thank you for the suggestion. We have incorporated additional details on the dedifferentiation of EE cells into ISCs via JAK-STAT signaling in P9 and expanded on how dedifferentiation contributes to tumorigenesis in P10.
- Line 194: Tissue regeneration therapies? Should elaborate more on this.
Thank you for the suggestion. We have added more about tissue regeneration therapies in P10.
- Inclusion of figures of results from discussed research articles would be good.
We added more figures to the manuscript.
- Lack of tables is another drawback. For example, table summarizing the various signaling pathways (e.g., Notch, Wnt, BMP, JAK-STAT, EGFR, etc.) involved in ISC regulation. It could list the pathway, its key components, and its role in ISC self-renewal, differentiation, or proliferation. A comparative table showing the different ISC regeneration mechanisms in Drosophila and mammals, including key processes like dedifferentiation, mitosis, and signaling activation.
We agree with the reviewer. We have added the figs to summarize the ligands, key components and their roles in ISC maintenance, proliferation and dedifferentiation in Fig. 2B, Fig. 3 and Fig. 5.
- Should include the limitations and future prospects as well.
Thank you for the suggestion. We have revised the conclusion as "Summary and Future Prospects," incorporating more details on the limitations of current studies using Drosophila and mouse models. Additionally, we expanded on future directions for intestinal stem cell research in P14.
- Conclusion should be rewritten. It should concisely summarize or highlight the information stated in the manuscript, rather than stating what was discussed in the above text.
We have revised the conclusion as "Summary and Future Prospects in P14-15.
- Only 11 out of 131 are from recent (on or after 2020) literature, should include more as the topic has several articles published in the recent past.
We have added more references on reviews after 2020, for example: 31, 33, 43, 137.
Remark
The review offers significant insights and manuscript is well organized and presented. Above suggestions should be included to enhance the relevance of the review.
Thank you for the positive feedback.
Reviewer 2 Report
Comments and Suggestions for Authors
Trubin and co-workers describe intestinal stem cell pool maintenance, stem cell regeneration, proliferation, and related signaling pathways. The article is well-written, clear, and provides adequate amount of interesting details to enrich the scientific field.
Corrections to be done: The review describes the intestinal stem cell biology in Drosophila melanogaster. It does not describe stem cell plasticity, so that word must be removed from the title. Because almost the entire article describes the D. melanogaster stem cell biology, the title must be changed accordingly, etc. "Intestinal Stem Cell Pool Maintenance in Drosophila Melanogaster", or similar. The title must contain the species D. melanogaster not to mislead the reader to think that the manuscript describes human stem cell biology.
In one of the figures, e.g., figure 3, the authors should add the ligands inducing the signaling.
Author Response
Trubin and co-workers describe intestinal stem cell pool maintenance, stem cell regeneration, proliferation, and related signaling pathways. The article is well-written, clear, and provides adequate amount of interesting details to enrich the scientific field.
Corrections to be done: The review describes the intestinal stem cell biology in Drosophila melanogaster. It does not describe stem cell plasticity, so that word must be removed from the title. Because almost the entire article describes the D. melanogaster stem cell biology, the title must be changed accordingly, etc. "Intestinal Stem Cell Pool Maintenance in Drosophila Melanogaster", or similar. The title must contain the species D. melanogaster not to mislead the reader to think that the manuscript describes human stem cell biology.
We agree with the reviewer’s suggestion and have revised the title to Regulation of the Intestinal Stem Cell Pool and Proliferation in Drosophila.
In one of the figures, e.g., figure 3, the authors should add the ligands inducing the signaling.
We thank the reviewer for the suggestion and add the ligands information to Fig. 2B and Fig. 3.
Reviewer 3 Report
Comments and Suggestions for Authors
Dear authors
These findings offer valuable insights into the essential role of ISCs in preserving normal tissue integrity and function. Disruptions in ISC pathways are implicated in a range of pathological conditions, notably CRC, where inflammation and epithelial dysregulation often exacerbate disease progression. A minor adjustment to the authors’ discussion could further emphasize the potential of targeting ISC pathways as a preventive strategy against CRC and related pathologies. By focusing on ISC pathway modulation, the authors could highlight how such interventions may play a role not only in managing ISC dysfunction but also in preventing cancer onset and disease pathogenesis.
Major reviews
1. Author added discussion highlights the critical nature of ISC maintenance and suggests potential therapeutic avenues focused on ISC health as a means to counter CRC.
2. The authors are encouraged to explore an alternative approach to further validate their findings and to emphasize more clearly the novelty of their results. Additionally, a more detailed explanation of the in vitro and in vivo models used, including the underlying rationale for selecting these methods, would enhance the clarity and relevance of the study's methodological framework

Author Response
Major reviews
- Author added discussion highlights the critical nature of ISC maintenance and suggests potential therapeutic avenues focused on ISC health as a means to counter CRC.
Thank you for the suggestion. We have revised the Summary and Future Prospects section to emphasize the importance of ISC maintenance and to highlight how disruptions in key pathways, such as the loss of Notch signaling, can contribute to tumor development. We also emphasize that misregulation of ISC activity can lead to gastrointestinal diseases and colorectal cancer (CRC).
- The authors are encouraged to explore an alternative approach to further validate their findings and to emphasize more clearly the novelty of their results. Additionally, a more detailed explanation of the in vitro and in vivo models used, including the underlying rationale for selecting these methods, would enhance the clarity and relevance of the study's methodological framework.
Most studies discussed in this review focus on in vivo models, but in vitro models, such as human organoids, are also essential for advancing tumorigenesis research and bringing the field closer to clinical applications. We have included this point in the Summary and Future Prospects section in P14.
Round 2
Reviewer 3 Report
Comments and Suggestions for Authors
Dear authors
These findings offer valuable insights into the essential role of ISCs in preserving normal tissue integrity and function. Disruptions in ISC pathways are implicated in a range of pathological conditions, notably CRC, where inflammation and epithelial dysregulation often exacerbate disease progression. A minor adjustment to the authors’ discussion could further emphasize the potential of targeting ISC pathways as a preventive strategy against CRC and related pathologies. By focusing on ISC pathway modulation, the authors could highlight how such interventions may play a role not only in managing ISC dysfunction but also in preventing cancer onset and disease pathogenesis.
Major reviews
1. Author added discussion highlights the critical nature of ISC maintenance and suggests potential therapeutic avenues focused on ISC health as a means to counter CRC.
Thank you for the suggestion. We have revised the Summary and Future Prospects section to emphasize the importance of ISC maintenance and to highlight how disruptions in key pathways, such as the loss of Notch signaling, can contribute to tumor development. We also emphasize that misregulation of ISC activity can lead to gastrointestinal diseases and colorectal cancer (CRC). >> Okay
2. The authors are encouraged to explore an alternative approach to further validate their findings and to emphasize more clearly the novelty of their results. Additionally, a more detailed explanation of the in vitro and in vivo models used, including the underlying rationale for selecting these methods, would enhance the clarity and relevance of the study's methodological framework
Most studies discussed in this review focus on in vivo models, but in vitro models, such as human organoids, are also essential for advancing tumorigenesis research and bringing the field closer to clinical applications. We have included this point in the Summary and Future Prospects section in P14. >> Okay
Second Revision
Dear Authors
Thank you for your effort in revising the manuscript according to my guidance. I appreciate the improvements made, though there are a few minor adjustments that would further enhance the clarity and professionalism of the content. I encourage you to consider these refinements to ensure the highest quality presentation.
Minor review
1. Page 5, figure 3 : I recommend that the authors revise the subtitles for clarity and add concise, informative figure legends to enhance the readability and flow of the manuscript. This will provide readers with a clearer understanding of each section and the data presented in the figures.
2. Page 4, figure 5: The authors are requested to revise the table titles (not figure titles) for improved clarity and accuracy in the manuscript.

The English could be improved to more clearly express the research.
Author Response
Minor review
- Page 5, figure 3 : I recommend that the authors revise the subtitles for clarity and add concise, informative figure legends to enhance the readability and flow of the manuscript. This will provide readers with a clearer understanding of each section and the data presented in the figures.
Thank you for the suggestion. We have revised the figure legend for Fig. 3.
“Figure 3. Ligands and cytokines produced in ISCs, EBs, ECs and visceral muscle activate JAK-STAT, EGFR, Wingless, BMP and Insulin signaling pathways to regulate ISC maintenance and proliferation.”
- Page 4, figure 5: The authors are requested to revise the table titles (not figure titles) for improved clarity and accuracy in the manuscript.
Thank you for the suggestion. We have updated the fig title with the table title and have replaced it in the manuscript.
“Table 5. Comparison of signaling network that regulates ISC pool, ISC proliferation and ISC regeneration in the Drosophila and mammalian intestines.”